# A Pair of Prognostic Biomarkers in Triple-Negative Breast Cancer: KLK10 and KLK11 mRNA Expression

**DOI:** 10.3390/life12101517

**Published:** 2022-09-28

**Authors:** Yueyang Liu, Weiwei Gong, Sarah Preis, Julia Dorn, Marion Kiechle, Ute Reuning, Viktor Magdolen, Tobias F. Dreyer

**Affiliations:** 1Clinical Research Unit, Department of Obstetrics and Gynecology, Technical University of Munich, 81675 Munich, Germany; 2Department of Gynecology, Guangdong Provincial People’s Hospital, Guangdong Academy of Medical Sciences, Guangzhou 519041, China; 3Department of Hematology/Oncology, Guangzhou Women and Children’s Medical Center, Guangzhou 519041, China

**Keywords:** triple-negative breast cancer, KLK10, KLK11, mRNA

## Abstract

Triple-negative breast cancer (TNBC) is an aggressive breast cancer subtype with poor patient prognosis and limited therapeutic options. A lack of prognostic biomarkers and therapeutic targets fuels the need for new approaches to tackle this severe disease. Extracellular matrix degradation, release, and modulation of the activity of growth factors/cytokines/chemokines, and the initiation of signaling pathways by extracellular proteolytic networks, have been identified as major processes in the carcinogenesis of breast cancer. Members of the kallikrein-related peptidase (KLK) family contribute to these tumor-relevant processes, and are associated with breast cancer progression and metastasis. In this study, the clinical relevance of mRNA expression of two members of this family, KLK10 and KLK11, has been evaluated in TNBC. For this, their expression levels were quantified in tumor tissue of a large, well-characterized patient cohort (*n* = 123) via qPCR. Although, in general, the overall expression of both factors are lower in tumor tissue of breast cancer patients (encompassing all subtypes) compared to normal tissue of healthy donors, in the TNBC subtype, expression is even increased. In our cohort, a significant, positive correlation between the expression levels of both KLKs was detected, indicating a coordinate expression mode of these proteases. Elevated KLK10 and KLK11 mRNA levels were associated with poor patient prognosis. Moreover, both factors were found to be independent of other established clinical factors such as age, lymph node status, or residual tumor mass, as determined by multivariable Cox regression analysis. Thus, both proteases, KLK10 and KLK11, may represent unfavorable prognostic factors for TNBC patients and, furthermore, appear as promising potential targets for therapy in TNBC.

## 1. Introduction

Triple-negative breast cancer (TNBC) is an aggressive subtype of breast cancer, characterized by poor patient prognosis [1,2,3]. Its molecular features exclude these patients from available targeted therapies, such as anti-hormonal treatment or anti-HER2-targeting agents [3,4], due to the lack of estrogen (ER) and progesterone receptor (PR) expression, as well as the absence of epidermal growth factor receptor 2 (HER2) overexpression. Therefore, this cancer phenotype strongly limits the available therapeutic options to systemic chemotherapies (such as platinum-, taxane-, or anthracycline-containing chemotherapy regimens or post-adjuvant capecitabine therapy), which, unfortunately, results in only minor changes in the patient outcome, while simultaneously enhancing severe side effects [5]. The overall successful implementation of PARP inhibitors in breast cancer therapy for patients with BRCA1/2 mutation also seems to be a promising regimen for TNBC treatment; however, only 20% of the TNBC patients carry the relevant genetic aberrations [6]. Moreover, additional antibody-based therapy approaches, such as immune check-point inhibition (pembrolizumab) [7] or antibody–drug conjugates (sacituzumab govitecan or trastuzumab duocarmazine) [8] may expand the therapeutic options. Still, valid biomarkers for the development of individualized targeted treatment are urgently needed to improve clinical TNBC management.

Many members of the kallikrein-related peptidase family, which encompasses 15 closely related serine proteases [9], are dysregulated and implicated in biologically relevant processes in various cancer types, including ovarian, breast, and prostate cancer [10]. Several KLKs are expressed in normal breast tissue, and most of them (KLK4–8 and KLK12) have been recognized as potential diagnostic and/or prognostic markers in breast cancer, particularly in TNBC [11,12,13,14,15]. Moreover, the dysregulated expression pattern of distinct KLKs during carcinogenesis strongly points towards a functional role of these proteases for breast cancer progression. Especially, in TNBC tissues, higher concentrations of distinct KLKs have been detected than in other breast cancer subtypes [12,13].

KLK10 is widely distributed among various human organs and biological fluids, and is broadly involved in pathophysiological processes [16]. Cumulative evidence indicates that KLK10 represents an unfavorable prognostic marker in colon, renal clear cell, colorectal, and gastric cancer [17,18,19]; however, in contrast, KLK10 has also been associated with tumor-suppression in prostate and testicular cancer due to its capacity to modulate cell proliferation and apoptosis [20]. Moreover, KLK10 turned out to be a favorable biomarker, e.g., in ovarian cancer [15,21]. KLK10 mRNA as well as antigen expression levels were found to be downregulated in breast cancer tissue, compared to non-cancerous breast tissue [22,23], however, the specific regulatory mechanism has not been clarified yet. In addition to the prognostic value and solid expression in breast cancer, KLK10 has been identified to be an important factor contributing to tamoxifen and trastuzumab resistance, suggesting it as a potential new therapeutic target [24,25].

KLK11 seems to be an important enzyme for several physiological processes in humans, and is expressed in the tissue of the esophagus, salivary gland, skin, vagina, and cervix [26]. Geng et al. [15,27] reported that both high KLK11 mRNA and protein levels are significantly associated with a favorable prognosis in high-grade serous ovarian cancer, implying that KLK11 may play an anti-tumorigenic role in this cancer subtype. A similar prognostic value has been described for laryngeal carcinoma, non-small lung cancer, and esophageal carcinoma [28,29,30]. However, there are also reports linking elevated KLK11 levels with poor prognosis in rectal and colorectal cancer [31,32]. Thus, the overall role of KLK11 in the progress of cancer remains inconsistent. For breast cancer, at least, transcriptional changes during disease progression have been described before. KLK11 expression levels are higher in low-grade than in high-grade breast cancer tissues and even a loss of KLK11 was described in highly malignant tissues [33]. Nevertheless, additional information on the potential tumor biological role of KLK11 in TNBC is lacking.

In addition to the individual effects of the KLKs, these proteases are integrated in proteolytic networks, and are often coordinately expressed and regulated. In fact, a KLK10 and KLK11 axis has been previously described in ovarian cancer. In the same study, coordinated expression of these two factors was described accompanied by an improved prognostic value for the combined parameters, as compared to the prognostic power of each individual factor [15,27].

Altogether, KLK10 and KLK11 may have the potential to serve as prognostic biomarkers, and represent possible targets for therapy in breast cancer. Thus, in the present study, we aimed at exploring the hypothesis that KLK10 and KLK11 may be coregulated in TNBC, and that their mRNA expression levels individually or in combination may re- present prognostic biomarkers.

## 2. Materials and Methods

### 2.1. Patients

For this study, 123 pathologically classified TNBC tissue specimens were analyzed. Surgery was performed at the Department of Obstetrics and Gynecology (*TU Munich*) between the years 1988 and 2012. Patients did not receive chemotherapy prior to surgery. Tumor tissue was stored in liquid nitrogen immediately after surgery and analyzed by a pathologist. The TNBC status was evaluated via standard clinical protocols, which include immunohistochemical staining for estrogen receptor (ER) and progesterone receptor (PR), as well as the human epidermal growth factor 2 (HER2). Tissue specimens with receptor staining below 1% as well as IHC score 0 or 1 for HER2 were considered to be TNBC. In case of HER2 staining with score 2, HER2 gene amplification was further secured via fluorescence in situ hybridization (FISH).

Within this cohort, the predominant breast cancer subtype was classified as invasive ductal, grade 2/3 (109/123); the remaining cases were less frequently occurring subtypes, such as medullary and lobular cancer, among others. All patients underwent standard surgical procedures, including mastectomy or breast-conserving surgery. None of them exhibited distant metastases at the time of surgery. Patients were treated with adjuvant anthracycline- or cyclophosphamide-based chemotherapy according to the consensus recommendations at that time. Chemotherapy was administered in 72% (89/123) of cases. The age of the patients ranged from 30 to 96 years (median, 55 years), and the tumor size ranged from 0.5 to 11 cm (median, 2.25 cm). Clinical follow-up data were available from 4 to 286 months for OS (median, 79 months), and from 4 to 269 months for DFS (median, 71 months). The study was conducted under the *Declaration of Helsinki* and was approved by the local Ethics Committee (TU Munich, No. 491/17 S). Written informed consent is available for all patients.

### 2.2. Real-Time Polymerase Chain Reaction

The quantitative polymerase chain reaction (qPCR) was performed applying a hydrolysis probe-based approach (for details see [34]). Gene-specific primers, as well as target-specific probes, were designed by using the Roche Universal Probe Library Assay Center software (KLK10: Forward: 5′-CAGGTCTCGCTCTTCAACG-3′ Reverse: 5′-GAGCCCACAGTGGCTTGT-3′ Probe: 5′-FAM-TCCACTGC-3′ KLK11: Forward: 5′-GCTTGCTCTGGCAACAGG-3′ Reverse: 5′-AGTGAGGCTTGCACTCGAAC-3′ Probe: 5′-FAM-GAGACCAG-3′ HRPT1: 5′-TGACCTTGATTTATTTTGCATACC-3′ Reverse: 5′-CGAGCAAGACGTTCAGTCCT-3′ Probe 5′-FAM-GCTGAGGA-3′). All primers detect transcript variants encoding the full-length KLK10 and KLK11 proteins, respectively. For quantification of gene expression, the 2^ΔΔCT^ method was applied [34]. Amplification efficiencies were evaluated for all used assays. KLK11 and HPRT1 efficiencies, as determined by standard dilution series, were similar, whereas those of KLK10 and HPRT1 differed. Therefore, in the latter case, the values were efficiency-corrected (for details see [27]).

### 2.3. Statistical Analysis

The correlation of mRNA expression levels of KLKs with clinicopathological parameters of the patients was assessed using the chi-square test. Survival analyses were performed by constructing Kaplan–Meier curves. The log-rank test was used to evaluate group differences in the survival functions. Associations of KLKs and clinical parameters with patients’ survival were additionally determined by univariate and multivariable Cox regression analysis, and expressed as hazard ratios (HR) with 95% confidence intervals (95% CI). The correlations between continuous variables of KLKs were examined using the Mann–Whitney U test and Spearman rank correlation (r_s_). Box plots were drawn to indicate differences. All calculations were performed with the SPSS statistical analysis software (version 20.0; SPSS Inc., Chicago, IL, USA). Values of *p* ≤ 0.05 were considered statistically significant. The TCGA breast cancer dataset was assessed and acquired via the Xena Browser homepage. For the analysis, the so-called *fragments per kilo base of transcript per million mapped fragments (FPKM) normalization* was used. A higher FPKM is an accurate surrogate for higher RNA expression. This approach takes into account the length of the gene, and guarantees a better exclusion of batch effects and technical difficulties during the analysis [35]. In order to rule out repeating counts, especially of shorter fragments, and guarantee a valid comparability between different datasets, the data were curated applying the RNA-Seq Maximum Estimator correction, RSEM [36].

## 3. Results

### 3.1. KLK10 and KLK11 Expression in Tumor Tissues of Triple-Negative Breast Cancer

The overall expression of KLK10 and KLK11 in TNBC in comparison to the other KLKs was assessed using the publicly available TCGA breast cancer dataset. Both KLK10 and KLK11 mRNA were among the highest expressed KLK mRNAs in the TNBC subtype (Figure 1A), whereby the KLK10 mRNA levels slightly exceeded those of KLK11. Moreover, the mRNA expression of both KLKs was generally downregulated in breast cancer tissue (encompassing all subtypes) compared to normal tissue, but was higher in TNBC compared to other breast cancer subtypes (Figure 1B,C). Moreover, in the case of KLK10, mRNA expression was even increased compared to normal breast tissue (Figure 1B). We quantified the KLK10 and KLK11 mRNA expression by an established qPCR assay in 123 TNBC cases. The KLK10 expression ranged from 0.00 to 10.53 (median 0.12), that of KLK11 ranged from 0.00 to 2.70 (median: 0.01). The mRNA expression was categorized into a low-expressing vs. a high-expressing group (KLK10: 50th percentile; KLK11: 67th percentile).

By applying Spearman correlation analysis, a strong positive correlation was observed between KLK10 and KLK11 (r_s_ = 0.722, *p* < 0.001) in our study cohort. A similar correlation was detected in the publicly available TCGA dataset (r_s_ = 0.744, *p* < 0.001). The association between the groups with low and high mRNA expression for both KLKs was in addition validated by box plot analysis, further indicating a coregulation of the KLKs in TNBC (Figure 1D,E). A high expression of KLK10 mRNA is mostly associated with a high expression of KLK11 mRNA, applying the above-mentioned cutoffs.

### 3.2. Association of KLK10 and KLK11 mRNA Expression Levels with Disease-Free (DFS) and Overall (OS) Survival and Clinical Parameters

First, the relative expression levels of KLK10 and KLK11 mRNA were analyzed with respect to their association with established clinical variables, including age, lymph node status, and tumor size. The detailed results are summarized in Table 1. There was no relation between KLK10 and KLK11 expression on the mRNA level and clinicopathological features of TNBC.

The same clinical parameters, as well as chemotherapy (yes vs. no), together with the KLK10 and KLK11 mRNA expression were analyzed via univariate Cox regression analysis to determine their prognostic relevance regarding the patient outcome in the TNBC cohort (Table 2).

Among the clinical variables, advanced age and a positive lymph node status indicated significantly shorter DFS and OS. For patients treated with adjuvant chemotherapy, distinctly improved DFS and OS were observed compared to untreated patients. Moreover, TNBC patients with a high KLK10 mRNA expression, according to the median cutoff, also showed a significantly increased risk for shorter OS together with a trend towards significance for shorter DFS. Similar effects were noticed for the patients with high KLK11 mRNA expression. Here, elevated KLK11 mRNA expression showed negative prognostic potential for DFS and a trend towards significance for OS. The combination of KLK10 and KLK11 mRNA (low/low vs. high and/or high) was significantly associated with both a higher probability of disease recurrence and overall survival.

The impact of the biological factors on patient survival was also confirmed by Kaplan–Meier estimation. As shown by the respective survival curves (Figure 2), high KLK10 mRNA levels were significantly associated with shortened DFS, whereas high KLK11 expression was notably associated with both worse DFS and OS. In case of the combination of both factors, a significant association with DFS and OS was also observed.

A multivariable Cox regression analysis was performed to evaluate the independence of the KLKs as prognostic factors in TNBC (Table 3). A multivariate base model was constructed, which included the following clinicopathological parameters: age, lymph node status, and tumor size. Under these multiparametric conditions, age was the only clinical parameter displaying predictive power.

After the establishment of the base model, the tumor-based biological factors were separately added to the model. Among the analyzed factors, KLK10 mRNA expression significantly contributed to the base model for DFS. Elevated KLK11 mRNA expression turned out to be an independent predictor of shortened OS. The combination of both KLK10 and KLK11 mRNA expression did not foster a more pronounced impact on the base model. The multiparametric analysis was also performed with an extended base model, including chemotherapy (yes vs. no, Appendix A). Here, KLK10 mRNA expression remained an independent unfavorable prognostic factor for OS.

## 4. Discussion

Remodeling of the extracellular matrix in the tumor environment by members of the kallikrein-related peptidase family has been described to be substantial for cancer development and progression. The KLK-mediated tumor-promoting effects are notably manifested by the differential KLK expression within the tumor tissue driving the aggressiveness and progression of the disease in several cancer entities such as prostate, ovarian, and breast cancer [9,10]. In the present study, we quantified mRNA expression levels of KLK10 and KLK11 in a well-defined cohort of patients afflicted with triple-negative breast cancer, and evaluated their potential as prognostic factors.

A previous study by Luo and colleagues [16] analyzing the prognostic potential of KLK10 in breast cancer disclosed a correlation between high KLK10 antigen levels in tumor tissue and a higher risk of disease recurrence as well as cancer-related death. A similar correlation was observed in the HER2-positive breast cancer subtype. In silico analysis of 400 HER2-positive breast cancer patients of the TCGA dataset revealed a positive association between the KLK10 mRNA expression and a worse patient outcome [25]. In the present study, we also confirm the prognostic relevance of KLK10 for the TNBC subtype. Here, elevated KLK10 mRNA expression levels were also associated with worse DFS. Interestingly, the overall KLK10 mRNA expression in breast cancer (encompassing all subtypes) in comparison to healthy tissue was lower, which is in line with the findings of other studies [12,13]. However, as shown in the present paper, KLK10 expression was significantly elevated in the TNBC subtype, not only compared to the other breast cancer subtypes, but also to healthy tissue. This particular upregulation of KLK10 expression in TNBC suggests a functional involvement of this protease in the tumor biology of this breast cancer subtype.

Concerning KLK11, the prognostic impact in breast cancer is still not completely understood. However, there are several studies on gastric, prostate, and ovarian cancer [15,27,29,37], which reported an association of high KLK11 expression in tumor tissue with a favorable patient prognosis. In prostate cancer, these findings were further validated by serum studies [38]. In breast cancer, KLK11 was identified as a marker for early stages since its expression was diminished or even lost in the course of disease progression [33]. Upregulation of KLK11 expression in breast cancer tissue vs. adjacent healthy tissue was found to be associated with the presence of the estrogen receptor [39]. Still, KLK11 expression is highest in TNBC among all breast cancer subtypes and, as shown in the present study, high KLK11 expression levels in TNBC patients were associated with a poor patient prognosis.

All in all, both KLK10 and KLK11 appear to exhibit tumor-promoting properties in some cancer entities. These findings for KLK10 and KLK11, together with their structural homology and similar functionality, may indicate a coordinate expression, or at least an interdependent regulation, of their expression levels. In fact, in the present study, we found a positive coregulation of KLK10 and KLK11 in TNBC tissue, a mechanism that has also been described in other cancer entities, such as ovarian cancer [15]. This coregulation might be based on several underlying mechanisms. At first, gene expression of most of the KLKs is prone to regulation by steroid hormones [40,41]. In TNBC, ER and PR are only, if at all, weakly expressed, but the presence of androgens has been described as relevant for KLK gene expression, especially in this breast cancer subtype [42]. Moreover, most of the KLKs are embedded in regulatory networks comprising other proteases and signaling molecules [43]. This can include the activation of proteases via cleavage of inactive proforms [44], or the involvement of secondary signaling molecules to proceed downstream signaling such as the activation of protease-activated receptors (PARs) [45]. Furthermore, the proteolytic activity of proteases is tightly regulated via natural inhibitors, which is also true for KLKs. KLK10 and KLK11 are both tryptic-like serine proteases, and are thereby prone to similar inhibitory mechanisms. Common natural inhibitors for KLKs are, among others, increased Zn^2+^ concentrations or proteinogenic inhibitors such as LEKTI [46]. Additionally, the proximity of the KLK10 and KLK11 genes within the KLK locus on chromosome 19 might allow simultaneous regulation already on a genetic level. Additionally, the similar substrate profile of KLK10 and KLK11, as determined by positional scanning [47], makes both proteases prone to concurrent feedback regulation. Coordinated expression and regulation of activity of both factors may well explain the very limited increase of the prognostic value upon the combination of KLK10 and KLK11 in our analysis.

So far, the evaluation of the prognostic value of both KLKs is only based on the here-studied well-characterized and homogenous in-house cohort, thus, calling for further validation by use of other comparable datasets. To this end, we aimed at verifying the described prognostic value of these factors via in silico analyses of various publicly available datasets, applying our predefined cutoffs. However, we were not able to comprehensively confirm our findings, since the meta-analysis of the most commonly used datasets revealed a great variance of prognostic values with hazard ratios indicating either an association with a favorable or an unfavorable patient outcome. This may be due to the data quality, sample size, or heterogeneity within the analyzed cohorts (Appendix A).

The pathophysiological mechanisms driving the tumorigenic properties of KLK10 and KLK11 have been addressed in several studies on various cancer types. Several findings underline the link of increased KLK levels with tumor progression. (a) The remodeling of the ECM is crucial for several tumor-promoting processes, such as invasion, migration, and vascularization [48]. Both KLK10 and KLK11 have been described to be involved in these processes, thereby contributing to worse patient prognosis [49]. (b) Most of the KLKs have been associated with gene signaling activity. A commonly discussed mechanism is the activation of PARs and the corresponding downstream signaling [45]. In fact, there have been several tumor-promoting pathways described before that may be targeted by PAR signaling [50,51]. (c) In addition to the modulation of the ECM, there are several other potential substrates for both KLK10 and KLK11 that might interfere with disease progression. Within this context, it is conceivable that the shedding of growth factors and cytokines could influence tumor growth itself and increase immune evasion [52,53,54]. The consensus of the above-mentioned mechanisms may well contribute to the here-described unfavorable prognostic values of both KLK10 and KLK11 in TNBC.

In addition to the actual tumor-promoting properties of KLK10/11-triggered effects, there might be also implications towards improved therapy responses, granting them additional predictive value to both KLKs. Several reports have linked increased KLK10 expression to an increased tolerance to cisplatin therapy [55,56,57] and resistance to anti-HER2 antibody-mediated therapy [25]. For KLK11 similar reports indicate a connection with oxaliplatin resistance [58].

The scientific substantiation of the predictive value of both KLKs still needs validation by further pro- or retrospective studies, of larger patient cohorts. Especially, the impact on resistance against new therapy approaches such as PARP-inhibition will be of tremendous interest [59]. In addition to the negative impact of KLK10 and KLK11 expression in TNBC on patient prognosis, the contradictory data from other cancer entities, such as ovarian cancer, cannot yet be explained. However, it is tempting to speculate that the diverse protein/substrate composition of the tumor-surrounding microenvironment in different tumor types provides substantially discriminating substrate portfolios for both proteases, and thereby generates individual signaling outcomes for either breast or ovarian cancer [60].

## 5. Conclusions

In this study, we explored the potential of KLK10 and KLK11 as biomarkers in TNBC. However, the underlying pathophysiological mechanisms sill remain rather elusive. Moreover, the translation of the prognostic potential of both proteases on the protein levels needs further clarification. Additionally, the predictive value of KLK10 and KLK11 requires further pro- and retrospective evaluation. However, the limited expression of both KLK10 and KLK11 in healthy tissue, together with the overall negative impact on tumor progression make both KLKs interesting therapeutic targets, in addition to their biomarker potential.

## Figures and Tables

**Figure 1 life-12-01517-f001:**
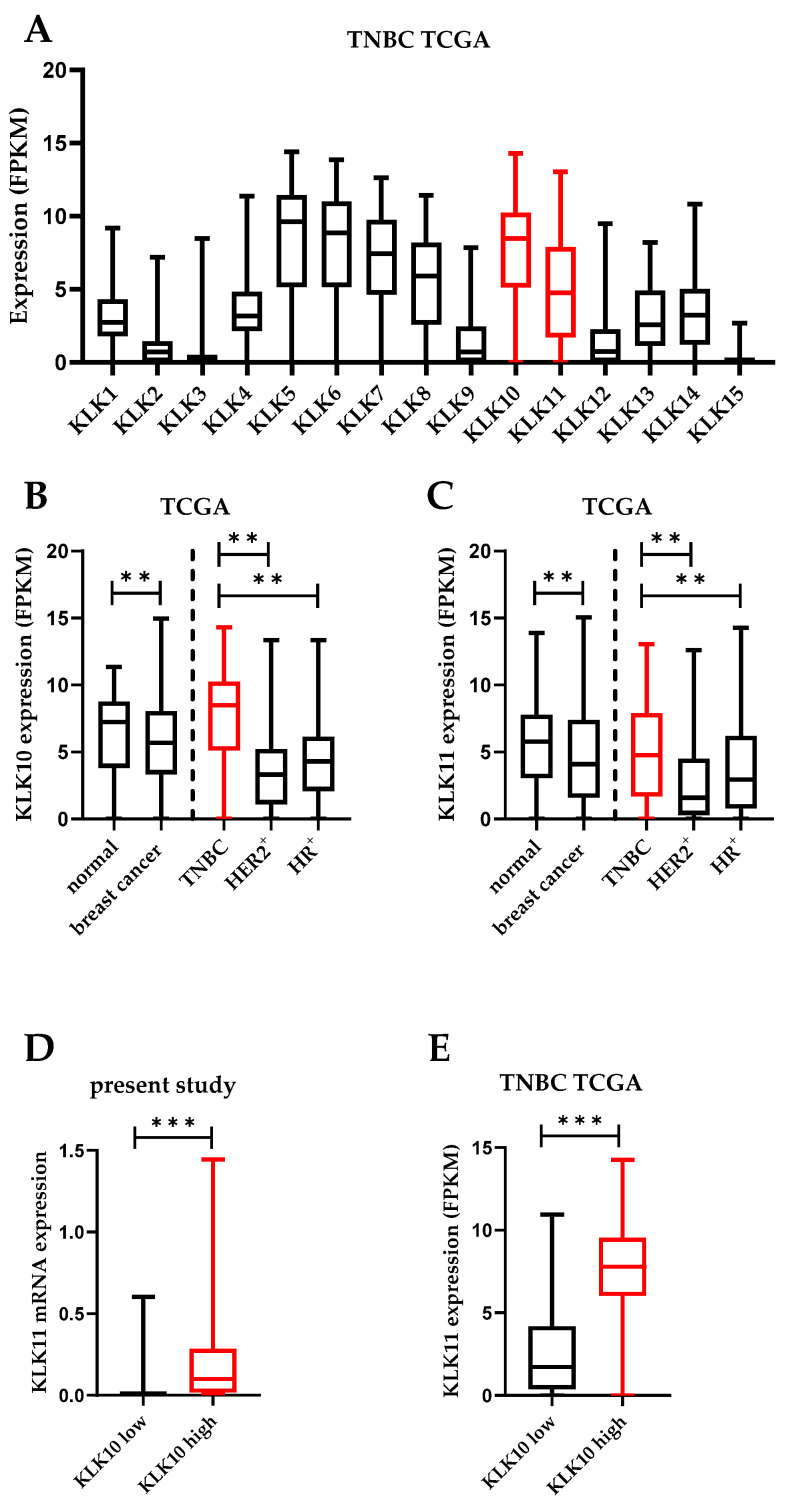
Expression of KLK10 and KLK11 mRNA in TNBC tissue. (**A**) Expression of the 15 kallikrein-related peptidases was assessed in TNBC tissue using the available data (*n* = 123) from the TCGA breast cancer dataset. KLK10 and KLK11 expression are highlighted in red. (**B**) and (**C**) In the same dataset, the differential expression between normal tissue (normal; *n* = 179), cancerous tissue (breast cancer; *n* = 1092) as well as subgroups thereof, triple-negative breast cancer (TNBC highlighted in red; *n* = 123), HER2 positive (HER2^+^; *n* = 102), and hormone-receptor-positive (HR^+^; *n* = 481) cancer, respectively, was analyzed for KLK10 (**B**) and KLK11 (**C**) expression. (**D**) Coordinate expression of KLK10 and KLK11 in our TNBC cohort (present study) was further validated in the TCGA dataset of TNBC cases (**E**). High KLK10 and KLK11 expression is highlighted in red. (Mann–Whitney U test) ** *p* > 0.005, *** *p* > 0.001.

**Figure 2 life-12-01517-f002:**
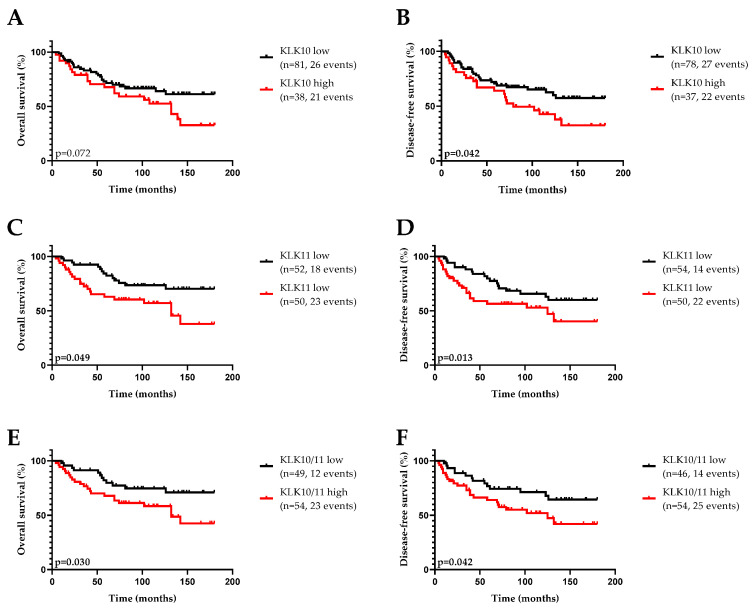
Probability of overall survival (OS) and disease-free survival (DFS) of patients with triple-negative breast cancer stratified by KLK10 and KLK11 mRNA expression levels in primary tumor tissue. Patients with elevated KLK11 (**C**) and KLK10 + KLK11 (**E**), but not KLK10 (**A**) mRNA expression levels show significantly shortened OS (Kaplan Meier analysis; KLK11, *p* = 0.049; KLK10/11, *p* = 0.030; KLK10, *p* = 0.072). Patients with higher mRNA levels of KLK10 (**B**), KLK11 (**D**), and the combination thereof (**F**) display significantly poor DFS (KLK10, *p* = 0.042; KLK11, *p* = 0.013; KLK10/11, *p* = 0.042), compared to those with low mRNA expression levels. Cutoff for KLK10 expression was the 50th percentile and for KLK11 expression the 67th percentile. The combined factors were dichotomized into low KLK10 and low KLK11 vs. high KLK10 and/or high KLK11 expression.

**Table 1 life-12-01517-t001:** Association between KLKs mRNA expression levels and clinicopathological parameters in patients with triple-negative breast cancer.

Clinicopathological Parameters	KLK10 ^a^	KLK11 ^a^	KLK10+11 ^a^
	Low/High	Low/High	Low/Others ^b^
**Age**	*p* = 0.294	*p* = 0.761	*p* = 0.390
≤60 years	43/25	33/28	26/35
>60 years	39/15	23/22	22/21
**Lymph node status**	*p* = 0.990	*p* = 0.473	*p* = 0.626
N0	45/22	33/26	28/30
N^+^	37/18	23/24	20/26
**Tumor Size**	*p* = 0.207	*p* = 0.665	*p* = 0.831
≤20 mm	25/8	15/15	14/15
>20 mm	56/32	41/34	34/40

^a^ Chi-square test (cutoff: KLK10 = 67th percentile, KLK11 = 50th percentile), ^b^ KLK10 low/KLK11 low versus KLK10 and/or KLK11 high. Due to missing values, numbers do not always add up to *n* = 123.

**Table 2 life-12-01517-t002:** Univariate Cox regression analysis of clinical outcome in triple-negative breast cancer for clinicopathological parameters and tumor biological factors.

Clinicopathological Parameters		DFS			OS	
	No ^a^	HR (95% CI) ^b^	*p*	No ^a^	HR (95% CI) ^b^	*p*
**Age**			**0.005**			**<0.001**
≤60 years	65	1		65	1	
>60 years	52	2.09 (1.25–3.52)		56	2.76 (1.62–4.70)	
**Lymph node status**			**0.023**			**0.018**
N0	63	1		65	1	
N^+^	54	1.82 (1.09–3.05)		56	1.86 (1.11–3.10)	
**Tumor Size**			*0.058*			*0.076*
≤20 mm	32	1		33	1	
>20 mm	84	1.89 (0.98–3.64)		87	1.85 (0.94–3.66)	
**KLK10 mRNA ^c^**			**0.045**			*0.076*
low	78	1		81	1	
high	37	1.78 (1.01–3.12)		38	1.68 (0.95–2.99)	
**KLK11 mRNA ^c^**			*0.053*			**0.016**
low	52	1		54	1	
high	50	1.85 (0.99–3.44)		50	2.29 (1.17–4.49)	
**KLK10+11 mRNA ^d^**			**0.046**			**0.035**
low	46	1		48	1	
high	54	1.95 (1.01–3.78)		54	2.12 (1.05–4.28)	
**Chemotherapy**			**<0.001**			**<0.001**
no	28	1		30	1	
yes	87	0.31 (0.172–0.546)		88	0.31 (0.173–0.562)	

Chi-square test, significant *p*-values (*p* < 0.05) are indicated in bold, trends towards significance (*p* < 0.08) in italics. ^a^ Number of patients. ^b^ HR: hazard ratio (CI: confidence interval) of univariate Cox regression analysis. ^c^ dichotomized into low and high levels by the 50th percentile for KLK11 and the 67th percentile for KLK10. ^d^ dichotomized into low levels by KLK10 low and KLK11 low and high levels by KLK10 high and/or KLK11 high.

**Table 3 life-12-01517-t003:** Multivariate Cox regression analysis of clinical patient outcome in triple-negative breast cancer for clinicopathological parameters and tumor biological factors.

Clinicopathological Parameters	DFS			OS		
	No ^a^	HR (95% CI) ^b^	*p*	No ^a^	HR (95% CI) ^b^	*p*
**Age**			**0.025**			**0.002**
≤60 years	54	1		54	1	
>60 years	41	1.61 (0.–4.09)		43	3.18 (1.53–6.59)	
**Lymph node status**			0.236			0.286
N0	51	1		53	1	
N^+^	44	1.49 (0.77–2.87)		44	1.46 (0.73–2.94)	
**Tumor Size**			0.305			0.423
≤20 mm	28	1		29	1	
>20 mm	67	1.51 (0.69–3.33)		68	1.41 (0.61–3.29)	
**KLK10 mRNA ^c^**			**0.019**			*0.058*
low	63	1		79	1	
high	32	2.19 (1.14–4.20)		36	1.95 (0.98–3.91)	
**KLK11 mRNA ^c^**			0.113			**0.044**
low	51	1		53	1	
high	44	1.70 (0.88–3.26)		44	2.06 (1.02–4.14)	
**KLK10+11 mRNA ^d^**			**0.049**			*0.054*
low	45	1		47	1	
high	50	1.97 (1.00–3.86)		50	2.03 (0.99–4.17)	

Chi-square test, significant *p*-values (*p* < 0.05) are indicated in bold, trends towards significance (*p* < 0.08) in italics. ^a^ Number of patients. ^b^ HR: hazard ratio (CI: confidence interval) of univariate Cox regression analysis. ^c^ dichotomized into low and high levels by the 50th percentile for KLK11 and the 67th percentile for KLK10. ^d^ dichotomized into low levels by KLK10 low and KLK11 low and high levels by KLK10 high and/or KLK11 high.

## Data Availability

The data that support the findings of this study are available on request from the corresponding author, [TD]. The data are not publicly available due to their containing information that could compromise the privacy of research participants. The data that support the findings of this study are openly available in https://xenabrowser.net/, reference number (TCGA Breast Cancer [61]).

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
