# Peer review of "A Pair of Prognostic Biomarkers in Triple-Negative Breast Cancer: KLK10 and KLK11 mRNA Expression"

_life, 2022, doi:10.3390/life12101517_

Round 1
Reviewer 1 Report
Does the use of adjuvant therapy in 89 of the 123 cases not interfere with the levels of KLK evaluated?
Citation 27 of qPCR materials and methods do not explain much of the applied "FPKM" (Fragments Per Kilo Base Transcript Per Million Fragments Mapped) normalization; it is suggested to explain more detail and add the meaning of the said acronym.
The concept of breast cancer is not clear in Figure 1. For example, was it an analysis of all types? Does it correspond to "n" = 123 of all cases? Similarly, in this figure, titles at the top of the graphs, such as "TGCA Breast Cancer Database" and "cohort," would help to understand the figure more easily.
Figure 2 needs to be revised; the legend does not match what is shown on the graphs, particularly when KLK10-KLK11 are displayed together (letter E).
Author Response
Point-by-point response to the questions raised (changes in the manuscript, except language editing, are highlighted in yellow):
Response to Reviewer 1 comments:
First of all, thank you very much for the valuable comments which helped to distinctly improve the quality of the manuscript.
Point 1: Does the use of adjuvant therapy in 89 of the 123 cases not interfere with the levels of KLK evaluated?
Response 1: The reviewer raised an important question, since the impact of chemotherapy on gene expression can indeed be strong. However, in our study, the analysis of the KLK mRNA expression was performed via qPCR in tumor tissue of the TNBC patients resected during the primary surgery. Therefore, at this time point, none of the patients had received a chemotherapy treatment. Thus, an interference of adjuvant treatment on the quantified mRNA levels is not possible. We now clarified this in the Material and Methods section as follows (page 4):
"For this study, 123, pathologically classified, TNBC tissue specimens were analyzed. Surgery was performed at the Department of Obstetrics and Gynecology between the years 1988 and 2012. Patients did not receive chemotherapy prior to surgery."
Point 2: Citation 27 of qPCR materials and methods do not explain much of the applied "FPKM" (Fragments Per Kilo Base Transcript Per Million Fragments Mapped) normalization; it is suggested to explain more detail and add the meaning of the said acronym.
Response 2: We agree that the normalization of the TCGA data has not been sufficiently explained in the manuscript leaving out some details which may be of potential interest to the scientific community. To this end, we now added a brief explanation of FPKM, including a more comprehensive citation in the statistics part of the manuscript. Furthermore, additional information on the normalization is given in this section, especially, concerning the comparability of the used data sets, as follows (page 5):
"For the analysis, the so-called fragments per kilo base of transcript per million mapped fragments (FPKM) normalization was used. A higher FPKM is an accurate surrogate for higher RNA expression. This approach takes into account the length of the gene and guarantees a better exclusion of batch effects and technical difficulties during the analysis [37]. In order to rule out repeating counts, especially of shorter fragments, and guarantee a valid comparability between different data sets, the data were curated applying the RNA-Seq Maximum Estimator correction RSEM [38]."
We have added two additional citations:
- Mortazavi, A et al. ;Nat. Methods 2008, 5 (7), 621. https://doi.org/10.1038/NMETH.1226.
- Li, B et al.; BMC Bioinformatics 2011, https://doi.org/10.1186/1471-2105-12-323.
Point 3: The concept of breast cancer is not clear in Figure 1. For example, was it an analysis of all types? Does it correspond to "n" = 123 of all cases? Similarly, in this figure, titles at the top of the graphs, such as "TGCA Breast Cancer Database" and "cohort," would help to understand the figure more easily.
Response 3: We thank the reviewer for this comment on the composition of the analyzed cohorts in Figure 1. The TCGA data set, we have been analyzing consists of 1,092 breast cancer cases (including all subtypes) and 179 healthy cases. We have performed the initial comparison of the KLK expression in the entire patient cohort (n=1,092) compared to the healthy data set. From the whole data, we have extracted those patients which have been classified as HER2+ (102 cases), HR+ (481 cases), and TNBC (123 cases), respectively. For the remaining cases of the data set, information concerning HR/HER2 receptor expression were missing. We have now clarified the above-mentioned distribution in the legend of Figure 1 as follows:
"In the same data base, the differential expression between normal tissue (normal; n=179), cancerous tissue (breast cancer; n=1,092) as well as subgroups thereof, triple-negative breast cancer (TNBC; n=123), HER2 positive (HER2+; n=102), and hormone receptor-positive (HR+; n=481) cancer, respectively, was analyzed for KLK10 (B) and KLK11 (C) expression."
In addition, we have now included titles above the graphs. Every panel now states the source of the data used for the analysis. This is either the TCGA data set or data from the present study cohort.
Point 4: Figure 2 needs to be revised; the legend does not match what is shown on the graphs, particularly when KLK10-KLK11 are displayed together (letter E).
Response 4: The reviewer is perfectly right and we have to sincerely apologize for accidentally including a preliminary version of Figure 2 into the originally submitted manuscript. The legend and the labeling within the figure panels of the now included final version of Figure 2 match each other.
Reviewer 2 Report
The issue could be of interest; however, I have some important remarks.
1. Authors should better describe the population of patients enrolled in this study; in particular, no data about pre/post-operative chemotherapy has been reported. Chemotherapy is an important prognostic factor that must be taken into account in uni-and multivariate analysis.
2. Considering the results of multivariate analysis, and the conflicting data about the role of kallikreins in cancer prognosis, I suggest to modify the Conclusions as follows:
In the abstract:
“Both proteases, KLK10 and KLK11, may represent unfavorable prognostic factors for TNBC patients”
In the main text:
“In this study, we have explored the potential of KLK10 and KLK11 as biomarkers in TNBC. The underlying pathophysiological mechanisms remain rather unknown, and also the translation of the prognostic potential on the protein levels needs to be further clarified. Additionally, the predictive value of KLK10 and KLK11 needs further pro- and retrospective evaluation”
3. Introduction
This section is too long and should be shortened.
Lines 44-48: “This phenotype strongly limits the available therapeutic options to systemic chemotherapies (like platinum- or taxane-containing chemotherapy regimens or post-adjuvant capecitabine therapy) which, unfortunately, results in only minor changes in the outcome for the price of higher burden in toxicity [5].” This phrase is not accurate; there is no mention of anthracyclines (a mainstay in TNBC chemotherapy) and new therapies such as immunotherapy or Ab-drug conjugates. I suggest authors to refer to more recent literature.
4. An accurate English revision is necessary
Author Response
Point-by-point response to the questions raised (changes in the manuscript, except language editing, are highlighted in yellow):
Response to Reviewer 2 comments:
We thank the reviewer for the helpful comments, which allowed us to significantly improve the overall quality of the manuscript. We also have cited and addressed the remarks in a point-to-point fashion:
Point 1: Authors should better describe the population of patients enrolled in this study; in particular, no data about pre/post-operative chemotherapy has been reported. Chemotherapy is an important prognostic factor that must be taken into account in uni-and multivariate analysis.
Response 1: The availability of a comprehensive population description is indeed important for the overall statement of the study. We have added additional clinical information for the patients in more detail. This includes age, tumor size, grading, and the range of the follow up-data. We also have clarified the time of sample collection. The tissue was collected and stored at the time of the surgery. At this time point, none of the patients had received chemotherapy. The concept of neoadjuvant treatment was rather rare at the time of the sample collection. The new paragraph now reads as follows (page 4):
"Within this cohort, the predominant breast cancer subtype was classified as invasive ductal, grade 2/3 (109/123); the remaining cases were less frequently occurring subtypes like medullary, lobular, and others. All patients underwent standard surgical procedures including mastectomy or breast-conserving surgery. None of them had distant metastases at the time of surgery. Patients were treated with adjuvant anthracycline- or cyclophosphamide-based chemotherapy according to the consensus recommendations at that time. Chemotherapy was administered in 72 % (89/123) of all cases. The age of the patients ranged from 30 to 96 years (median, 55 years) and the tumor size ranged from 0.5 to 11 cm (median, 22.5 cm). Clinical follow up data were available from 4 to 286 months for OS (median, 79 months) and from 4 to 269 months for DFS (median, 71 months)."
The impact of chemotherapy, especially in multivariate analysis, is an interesting addendum to this study. We have performed the suggested analysis and have included chemotherapy treatment (yes vs. no) in the univariate (Table 2) and multivariate analysis (Table S1). As stated by the reviewer, treatment with chemotherapy is a strong prognostic factor. However, the expression of KLK10 mRNA remains a significant predictor for shorter overall survival in the multivariate analysis. The additional results are now mentioned in the main text (page 9 and 11, respectively). Moreover, we have added an additional table to the supplementary section including the whole analysis (Table S1):
"The same clinical parameters as well as chemotherapy (yes vs. no) together with the KLK10 and KLK11 mRNA expression were analyzed via univariate Cox regression analysis to determine their prognostic relevance regarding the patient outcome in the TNBC cohort (Table 2)."
"The multi-parametric analysis was also performed with an extended base model, including chemotherapy (yes vs. no, Supp. Table 1). Here, KLK10 mRNA expression remained an independent unfavorable prognostic factor for OS."
Point 2: Considering the results of multivariate analysis, and the conflicting data about the role of kallikreins in cancer prognosis, I suggest to modify the Conclusions as follows:
In the abstract:
“Both proteases, KLK10 and KLK11, may represent unfavorable prognostic factors for TNBC patients”
In the main text:
“In this study, we have explored the potential of KLK10 and KLK11 as biomarkers in TNBC. The underlying pathophysiological mechanisms remain rather unknown, and also the translation of the prognostic potential on the protein levels needs to be further clarified. Additionally, the predictive value of KLK10 and KLK11 needs further pro- and retrospective evaluation”
Response 2: We agree that a more cautious wording better represents the results of the study. We have changed the wording of both, the abstract (page 1) and conclusion (page 14), according to the Reviewer's suggestion.
“In this study, we have explored the potential of KLK10 and KLK11 as biomarkers in TNBC. However, the underlying pathophysiological mechanisms still remain rather elusive. Also, the translation of the prognostic potential of both proteases on the protein levels needs further clarification. Additionally, the predictive value of KLK10 and KLK11 requires further pro- and retrospective evaluation.”
Point 3: Introduction
This section is too long and should be shortened.
Response 3: According to the Reviewer's suggestion, we have substantially shortened the introduction (pages 1-3). The deleted phrases are also marked in yellow in the main text. During this editing process, we have also removed the following citations from the manuscript.
- Walker et al.;Int. J. Mol. Sci. 2018, 19 (10). https://doi.org/10.3390/IJMS19103028.
- Borgoño et al.; Nat. Rev. Cancer 2004, 4 (11), 876–890. https://doi.org/10.1038/NRC1474.
- Shaw et al.; Clin. Chem. 2007, 53 (8), 1423–1432. https://doi.org/10.1373/clinchem.2007.088104.
- Yousef et al:; Biochem. Biophys. Res. Commun. 2000, 276 (1), 125–133. https://doi.org/10.1006/bbrc.2000.3448.
- Chow et al:; Biol. Chem. 2008, 389 (6), 731–738. https://doi.org/10.1515/BC.2008.071/HTML.
Lines 44-48: “This phenotype strongly limits the available therapeutic options to systemic chemotherapies (like platinum- or taxane-containing chemotherapy regimens or post-adjuvant capecitabine therapy) which, unfortunately, results in only minor changes in the outcome for the price of higher burden in toxicity [5].”
This phrase is not accurate; there is no mention of anthracyclines (a mainstay in TNBC chemotherapy) and new therapies such as immunotherapy or Ab-drug conjugates. I suggest authors to refer to more recent literature.
The reviewer has correctly stated that the described therapy in the introduction is lacking some more (recent) important therapy approaches. We have overworked this section and added recent therapy approaches, including immunotherapies (e.g. immune check-point inhibition) and antibody-drug conjugates (trastuzumab and sacituzumab) (page 2):
“Therefore, this cancer phenotype strongly limits the available therapeutic options to systemic chemotherapies (like platinum-, taxane- or anthracyclin-containing chemotherapy regimens or post-adjuvant capecitabine therapy) which, unfortunately, results in only minor changes in the patient outcome while simultaneously enhancing severe side effects [5].“
"Moreover, additional antibody-based therapy approaches, such as immune check-point inhibition (pembrolizumab) [7] or antibody-drug conjugates (sacituzumab-govtecan or trastuzmab-duocarmazine) [8]."
For this, we have also added two more citations:
- Cortes et al.; Lancet 2020, 396 (10265), 1817–1828. https://doi.org/10.1016/S0140-6736(20)32531-
- Koster et al; Explor Target Antitumor Ther. 2022, 27-36, https://doi.org/10.37349/etat.2022.00069
Point 4: An accurate English revision is necessary
Response 4: We have revised the entire manuscript with respect to language and style.
Round 2
Reviewer 2 Report
Authors have revised/modified the text according with the reviewer’s suggestions